# Microbiological Activity during Co-Composting of Food and Agricultural Waste for Soil Amendment

**Vladimir Mironov** * 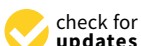, **Anna Vanteeva** 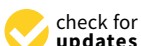 **and Alexander Merkel**

Winogradsky Institute of Microbiology, Research Center of Biotechnology, Russian Academy of Sciences, 119071 Moscow, Russia; very-well1966@mail.ru (A.V.); alexandrmerkel@gmail.com (A.M.)
* Correspondence: 7390530@gmail.com; Tel.: +7-(903)-7390530

**Abstract:** This study aims to establish the relationship between ambient parameters and the diversity, composition, and function of microbial communities that predominate at each stage of the co-composting of food and agricultural waste. Culture-based and culture-independent methods were used to investigate the changes in the microbiota. The favorable conditions of high initial humidity and C/N ratio caused a decrease in the richness and biodiversity of the microbiota when such conditions existed. During the thermophilic stage, the total microbial number increased, and active mineralization of organic matter was carried out by members of the genera *Bacillus*, *Caldibacillus*, *Aspergillus*, and *Penicillium*. The fungal community was sensitive to drastic temperature changes. *Byssochlamys* dominated among fungi during the transition from the mesophilic to the thermophilic stage and during cooling. The biodiversity increased with time and was associated with the dynamics of germination and nitrification indices, so that the more diverse the microbial community, the higher the properties of compost that stimulate plant growth and development. The microbial community of the mature compost, together with mineral plant nutrients ready for consumption and humic compounds, make this compost a good soil additive.

**Keywords:** composting; agricultural and food waste; microbial communities; thermophilic micro-organisms; experimental system

## 1. Introduction

Food wastes (FW) are formed along the whole chain of food supply, from agricultural production to home consumption [1]. Co-composting with more than one kind of raw material is a reliable technology for FW processing [2]. This is an intensive microbial process, which results in the biodegradation of the raw material for further humification and results in the production of compost, a product that improves the physical, chemical, and microbiological properties of soil [3–5]. Successful composting depends on a number of factors that affect microbial activity either directly or indirectly [4,6,7]: the origin and composition of the initial substrate, composting conditions, and inoculation with specific micro-organisms.

FW composition is highly diverse and depends on a number of factors, including the time of year (season), consumer habits, etc. The average water content in FW (87.9%) does not favor composting, because high humidity may result in the development of anoxic conditions [8]. Wheat straw may be used as a moisture-absorbing material in regions with developed agriculture [9]. At the same time, adding straw to FW results in a significant increase of the C/N ratio above its optimal values of 25–30 [10].

This high ratio is considered to result in nitrogen immobilization, low compost quality, and postponed biodegradation due to the high lignocellulose content of straw [11]. Some authors, however, have obtained positive results with other C/N values, e.g., 40 [12,13]. Because there are no experimental data on the biological and physicochemical processes occurring during co-composting of a mixture of FW and straw with an increased C/N ratio and water content, further experimental research is required.

Compost's stability, maturity and, therefore, agrochemical properties depend on the results of microbial activity [2]. Different microbial communities develop at different stages of composting [14]. One of the major trends is the investigation of microbial populations developing in the course of composting not only by traditional microbiological methods but also by using new-generation technologies, e.g., high-throughput sequencing [3].

These tools are used for identification and isolation (if possible) of the most significant microorganisms (degraders, nitrogen-fixing bacteria, denitrifying bacteria, and archaea, as well as bacteria solubilizing phosphorus and potassium). Greater understanding of microbial processes will make it possible to establish the optimal conditions for the growth of specific microorganisms at each stage of composting, thus, resulting in the production of an individual compost with improved nutrient properties for specific crops [2,3].

Various factors affecting the formation of compost microbial communities are known [15–18]: composting conditions, properties of the initial substrate and its microbiota, and changes in the physicochemical parameters in the course of the process.

While relations between bacterial [16,18,19] and bacterial–fungal [15,17] consortia and environmental factors for various substrates have been considered previously, the data on the effect of high initial values of C/N and water contents on the subsequent microbial activity and diversity are scarce, as are those on their dependence on the changing ambient conditions at all stages of co-composting FW and agricultural waste.

We suggested that the increase in content of carbon by adding monosubstrate (straw) should lead to decrease in diversity of microbial communities but not to decrease in the number of microorganisms and the rate of biodegradation. We also expected the mineralization of organic nitrogen and easily degradable organic carbon to occur mainly during the thermophilic stage of the process due to higher rates of these processes at elevated temperatures.

The goals of the present work were: (i) to investigate dynamic changes in the diversity and functions of microbial communities predominant at each stage of co-composting the mixed food and agricultural waste; (ii) to assess the relations between the community diversity and composition and the changing ambient conditions; and (iii) to reveal the main genera of microbial degraders. The study of co-composting was carried out using a mixture of food and agricultural waste at high initial values of humidity and C/N.

## 2. Materials and Methods

### 2.1. Compost Mixture

Composting was carried out using a model substrate containing 85% FW and 15% agricultural waste (wheat straw) broken down into the following ratio % (wt/wt): potato, 23.23; cabbage 23.23; apples, 9.31; oranges, 9.31; bananas, 9.31; ground meat, 2.65; ground fish, 1.06; bread, 5.32; cheese, 1.06; chicken eggs, 0.53; and wheat straw, 15.00. The presence of moisture-absorbing materials (grain crops, straw, sawdust, or peat) in the mixture with FW in the 15:85 ratio (raw weight) may efficiently prevent the undesired consequences of composting [20].

The raw materials were obtained from the Bunyatino Agricultural Farm and the Grunt Eco Industrial Composting Complex (Moscow region, Russia). The components were homogenized into fractions below 10 mm on an IK-07E fodder shredder (Avtomash, Russia) and mixed thoroughly in a BSE-63 concrete mixing machine (Kalibr, Russia). The initial physicochemical parameters of the substrate are listed in Table 1.

**Table 1.** The physicochemical parameters of mixed food and agricultural waste with an 85:15 raw weight ratio.

| Parameters | Units | Value * | Optimal Limits |
|---|---|---|---|
| pH | pH units | $6.9 \pm 0.4$ | 6.5–8.0 [14] |
| Electrical conductivity (EC) | $\mu S\,cm^{-1}$ | $434 \pm 21$ | |
| Water content | % | $72.6 \pm 1.2$ | 50–60 [14] |
| Kjeldahl total nitrogen (N) | % | $1.08 \pm 0.20$ | |
| Ammonium nitrogen (N-NH$_4$) | $mg\,kg^{-1}$ | $490 \pm 32$ | |
| Nitrate nitrogen (N-NO$_3$) | $mg\,kg^{-1}$ | $0.2 \pm 0.1$ | |
| Organic matter (OM) | % | $87.16 \pm 1.2$ | |
| Total content of organic carbon (C) | % | $48.42 \pm 1.2$ | |
| C/N ratio | | 44.8 | 25–30 [10,14] |
| Germination index (GI) | % | $69 \pm 8$ | |

* Data expressed as mean $\pm$ standard deviation ($n = 3$).

## 2.2. Experimental Setup

Composting was carried out for 98 days under laboratory conditions using an experimental setup of a six-chamber bioreactor for the aerobic solid-state biodegradation of organic waste (working volume, 150 dm$^3$), Figure 1.

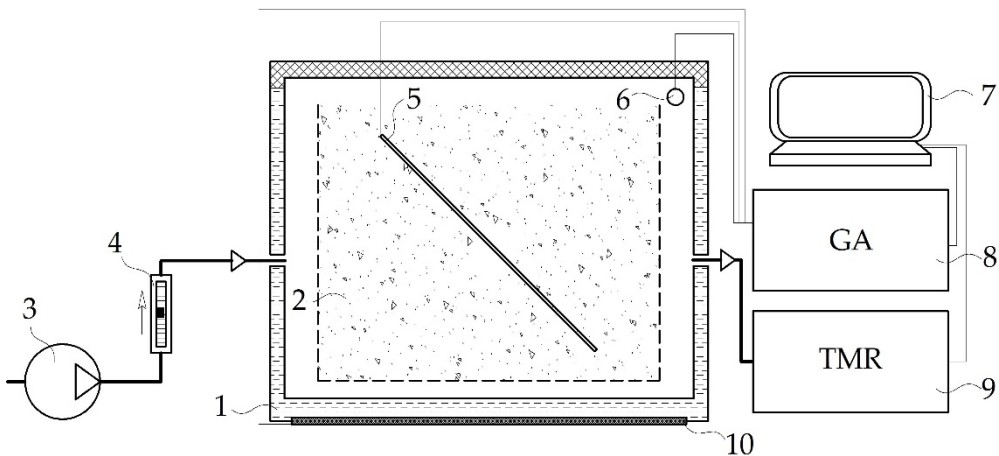

**Figure 1.** Schematic image of a laboratory bioreactor for the aerobic solid-state biodegradation of organic waste: composting chamber (1); compost mixture (2); compressor (3); air flow regulator (rotameter) (4); temperature sensors (5,6); computer (7); gas analyzer (GA) (8); temperature meter-regulator (TMR) (9); and heating element (10).

After 98 days, stabilized compost from all chambers was loaded onto a concrete platform and mixed mechanically to form a heap ~60 dm$^3$. To prevent drying, the compost was covered with a nonwoven geotextile material. The compost was preserved in this state for 10 months at $18.4 \pm 4.2$ °C. A similar composting time for kitchen and garden waste was successfully used in previous research [21]. The parameters of the process are listed in Table 2.

**Table 2.** The parameters for the composting process.

| Parameters | In-Vessel | Heap |
|---|---|---|
| Substrate volume (mass) per experiment | $2 \times 10$ dm$^3$ (12 kg) | Combined, ~60 dm$^3$ |
| Free space | 15 dm$^3$ | - |
| Number of replicates | Three simultaneously | Three simultaneously |
| Composting time | 98 days | 10 months |
| Process stages | Mesophilic, thermophilic, and cooling | Maturation |
| Aeration | Active, constant, 0.04 L min$^{-1}$ kg$^{-1}$ | Passive |
| Gas mixture analysis | In the chamber, once a day | None |
| Gas, range, error | CO$_2$ (from 0 to $10 \pm 0.1$ vol%) NH$_3$ (from 0 to $20 \pm 4$ mg m$^{-3}$) H$_2$S (from 0 to $10 \pm 2$ mg m$^{-3}$) | None |
| Substrate temperature measurement | Constant, from 0 to 100 °C | None |
| Ambient temperature | $24.3 \pm 1.5$ °C | $18.4 \pm 4.2$ °C |
| Substrate stirring (for 2 min) | Days 7, 14, 21, 28, and 56 | Days 98 and 389 |
| Addition of water (1 L) | Days 28 and 56 | None |

The experimental setup was in accordance with the previously reported system [22] and was operated as follows. Containers with the substrate (10 dm$^3$) were placed into each chamber. The containers were made of a polymer material with 5-mm perforations in the walls and bottom to provide aeration. Tap water was added to the substrate to compensate for evaporation-caused losses, so that the relative humidity was maintained from 53% to 69%.

The temperature in each chamber was maintained independently at the current substrate temperature $\pm 0.2$ °C using a heating element and an IVTM-7/2S temperature regulator (Eksis, Russia).

The substrate was aerated for 98 days with air of ambient temperature and consumption of 0.04 L min$^{-1}$ kg$^{-1}$ dry OM using a compressor and RMS-A-0.035 GUZ-2 rotameters (Pribor-M, Russia). The aeration rate was accepted based on the optimal value for the simultaneous control of released ammonium and volatile sulfur compounds during FW composting in aerated systems [22–24]. During days 98–389 of incubation, compost in the heap was aerated passively through the sides. The gas composition of the substrate atmosphere was determined daily with an MAG-6 S-1 gas analyzer (Eksis, Russia): CO$_2$, NH$_3$, and H$_2$S. The temperature mode in the substrate was monitored continuously.

### 2.3. Physicochemical Studies

Samples for physicochemical and microbiological analyses were collected on the first day of the experiment and then on days 7, 14, 21, and 28 (active phase of the process [10]); days 56 and 98, and at day 389 (mature compost). Samples for molecular genetic studies were stored at $-20$ °C.

The physicochemical parameters determined for all the samples used the generally accepted and modified procedures. The water-soluble compounds, pH, EC, and GI were determined from a suspension of the sample (10 g) in 300 mL distilled water [25]. Measurements of pH and EC were carried out on an ANION 4150 laboratory analyzer (Ifraspak-Analit, Russia). The water content (%) was determined by the thermogravimetric method on an EVLAS-2M humidity analyzer (Sibagropribor, Russia).

The OM content was determined using the thermogravimetric method at 430 °C [26] using a PM-16M-1200 muffle furnace (EVS, Russia). The total content of organic carbon (C,

%) was estimated as OM:1.8 [21]. The soluble ammonium (N-NH$_4$) and nitrate nitrogen (N-NO$_3$) [10,27] were determined on a Hach Lange DR 5000 spectrophotometer (Germany) according to the manufacturer's manual: ammonium was measured with the Nessler's reagent, and nitrate was measured by the method based on cadmium reduction.

The C/N ratio was calculated using experimental data. The effect of compost on plant growth was assessed by seed germination and the root length of *Raphanus sativus* based on the germination index (GI, %) calculation [28]. The Kjeldahl total nitrogen (N, %) in the dry matter was determined by sample mineralization in a DKL 6 automatic digester (Velp Scientifica, Italy) under conditions of heating up to 420 °C with concentrated sulfuric acid in the presence of hydrogen peroxide and a mixed catalyst with subsequent distillation of ammonium into a boric acid solution using an UDK 139 semiautomatic distillation system (Velp Scientifica, Italy) and titration with hydrochloric acid on an Easy Plus titrator, model Easy pH METTLER TOLEDO (Switzerland).

The humus group composition in mature compost was determined by the rapid pyrophosphate method with gravimetric evaluation [29]. The mass ratios of phosphorus and potassium in mature compost were determined using a method based on dry compost mineralization by heating with concentrated sulfuric acid in the presence of hydrogen peroxide. The optical density of the colored phosphorus–molybdenum complex was measured on a KFK-3 KM spectrophotometer (Tekhnokom, Russia). The total potassium was determined by photometry of the potassium radiation on an AA-7000 atomic absorption spectrophotometer (Shimadzu, Japan).

### 2.4. Microbiological Studies

For microbiological analysis, the samples (1 g) were resuspended in 10 mL of sterile tap water with glass beads and shaken on a Bio Vortex V1 (Biosan, Latvia) for 30 s. Serial tenfold dilutions of this material were then plated on sterile growth media (the composition is listed below) with subsequent incubation for 120 h at 28 and 50 °C.

The sanitary microbiological parameters of mature compost were studied using generally accepted methods [30]. The total microbial number (TMN) of heterotrophic aerobic microorganisms was determined by plating tenfold dilutions of the compost suspension onto a rich agar medium. The enriched clostridial medium contained the following (g L$^{-1}$): yeast extract, 13; peptone, 10; NaCl, 5; sodium acetate, 3; glucose, 5; starch, 1; cysteine hydrochloride, 0.5; and pH 7.0.

Analysis of the physiological groups of aerobic micro-organisms was carried out on selective solid media. Proteolytics were isolated on the modified Eikman agar (g L$^{-1}$): peptone, 5; yeast extract, 3; dry skim milk, 1; agar, 15; and pH 7. Amylolytics were grown on the following medium (g L$^{-1}$): soluble starch, 10; K$_2$HPO$_4$, 1; MgSO$_4$, 1; NaCl, 1; (NH$_4$)$_2$SO$_4$, 2; CaCO$_3$, 2; FeSO$_4$, 0.001; MnCl$_2$, 0.001; agar, 20; and pH 7.2.

The medium for ligninolytic micro-organisms [31] contained the following (g L$^{-1}$): tannin, 5; NaNO$_3$, 3; K$_2$HPO$_4$, 1; MgSO$_4$ · 7H$_2$O, 0.5; KCl, 0.5; agar, 30. Hutchinson–Clayton medium was used to isolate cellulolytics (g L$^{-1}$): microcrystalline cellulose, 5; NaNO$_3$, 2.5; K$_2$HPO$_4$, 1; MgSO$_4$ · 7H$_2$O, 0.3; NaCl, 0.1; CaCl$_2$ · 4H$_2$O, 0.1; FeCl$_3$ · 6H$_2$O, 0.01; agar, 20; and pH 7.2.

Phosphate-solubilizing micro-organisms were revealed on the Pikovskaya medium (g L$^{-1}$): yeast extract, 0.5; glucose, 10, Ca$_3$(PO$_4$)2, 5; (NH$_4$)$_2$SO$_4$, 0.5; KCl, 0.2; MgSO$_4$, 0.1; MnSO$_4$, 0.0001; FeSO$_4$, 0.0001; agar, 30; and pH 7.0. Nitrogen-fixers were grown on modified Ashby medium (g L$^{-1}$): sucrose, 20; K$_2$HPO$_4$, 0.2; MgSO$_4$, 0.2; NaCl, 0.2; K$_2$SO$_4$, 0.1; CaCO$_3$, 5; Fedorov trace elements solution, 1 mL; and agar, 17.

Winogradsky media was used for the isolation of stage I and stage II nitrifying bacteria contained, respectively (g L$^{-1}$): (NH$_4$)$_2$SO$_4$, 2; K$_2$HPO$_4$, 1; MgSO$_4$ · 7H$_2$O, 0.5; NaCl, 2; FeSO$_4$ · 7H$_2$O, 0.05; CaCO$_3$, 5; and NaNO$_2$, 1; K$_2$HPO$_4$, 0.5; MgSO$_4$ · 7H$_2$O, 0.5; NaCl, 0.5; FeSO$_4$ · 7H$_2$O, 0.4; and Na$_2$CO$_3$, 1. As ligninolytic micro-organisms are known to commence active growth at the cooling and maturation phases [32], the abundance of ligninolytic organisms was determined on day 28 and later.

After incubation, the colony-forming units (CFU) on agar media were enumerated. The growth and activity of nitrifying bacteria cultivated in liquid media was monitored on a Hach Lange DR 5000 spectrophotometer (Germany) according to the concentrations of the substrates (N-NH$_4$ and N-NO$_2$): the former was measured with Nessler's reagent, and the latter, by a method based on the nitrite interaction with iron sulfate (according to the Hach Lange DR 5000 manual).

### 2.5. Profiling of Prokaryotic and Fungal Communities Based on 16S rRNA Gene and Internal Transcribed Spacer (ITS)

The composition of the microbial community was analyzed by the number of 16S rRNA genes and ITS copies. DNA was isolated from the compost samples using the commercial FastDNA SpinKit (MP Bio, Salt Lake City, UT, USA) according to the manufacturer's instructions.

Libraries of the V4 region of the 16S rRNA gene for Illumina MiSeq high-throughput sequencing were prepared according to the protocol [33]. The following primer system was used to prepare the amplicons: forward (5′-CAAGCAGAAGACGGCATACGA GATGTGACTGGAGTTCAGACGTGTGCTCTTCCGATCT XXXXXX ZZZZ GTGBCAGC MGCCGCGGTAA-3′), containing, respectively, the 5′ Illumina Linker Sequence, Index 1, Heterogeneity Spacer [34], and 515F primer sequence [35]; and the reverse primer (5′-AATGATACGGCGACCACCGAGATCTACACTCTTTCCCTACACGACGCTCTTCCGATCT XXXXXX ZZZZ GACTACNVGGGTMTCTAATCC-3′), containing the 3′ Illumina Linker Sequence, Index 2, Heterogeneity Space, and the Pro-mod-805R primer sequence [36], respectively.

The ITS libraries were prepared in a similar way, using the primer system ITS86F (F) 5′-GTGAATCATCGAATCTTTGAA-3′ [37]–ITS4 (R) 5′-TCCTCCGCTTATTGATATGC-3′ [38] at the 3′ termini, upstream of the described oligonucleotide constructs. For each DNA sample, two libraries were prepared, which were sequenced in parallel using the MiSeq Reagent Micro Kit v2 (300-cycles) MS-103-1002 (Illumina, San Diego, CA, USA) on a MiSeq sequencer (Illumina, San Diego, CA, USA) according to the manufacturer's recommendations.

The primary processing of the raw reads was carried out as described earlier [39]. All 16S rRNA gene sequence reads were then processed by the SILVAngs 1.3 pipeline [40] using the default settings: 98% similarity threshold was used for creating operation taxonomic units (OTUs) tables; 93% was the minimal similarity to the closest relative that was used for classification (other reads were assigned as "No Relative"). All ITS sequence reads were processed by the Knomics-Biota [41] using the "ITS fungi" pipeline, where the reads were classified by mapping against the UNITE database version 7.2 (QIIME release, version 01.12.2017) using the BWA-MEM algorithm (BWA version 0.7.12-r1039). A 97% similarity threshold was used for creating OTUs.

### 2.6. Statistical Analysis

Throughout the study, three replicates of each treatment were used. The data were subjected to analysis of variance (ANOVA) using the least significance difference test and comparing the difference between the specific treatments. The test of significance was determined at $p < 0.05$. Error bars are the mean $\pm$ standard deviation of three replicates. The method of dimension reduction was principal coordinate analysis (PCoA); dissimilarity metric: weighted UniFrac [42]. The Pearson correlation coefficient (r) was used to determine the relationships between different variables. The Chao1 richness index [43] and the Shannon and Simpson biodiversity indices were calculated.

## 3. Results and Discussion

*3.1. Dynamics of the Physicochemical Parameters of Composting Mixed Food and Agricultural Waste*

3.1.1. Temperature and $CO_2$ Production

The temperature increased to 40 °C during the first 16 h. The heat generation then slightly decreased, and the local maximum of 44 °C was reached after 48 h, when the heat generation rate was equal to the heat loss into the environment, Figure 2a.

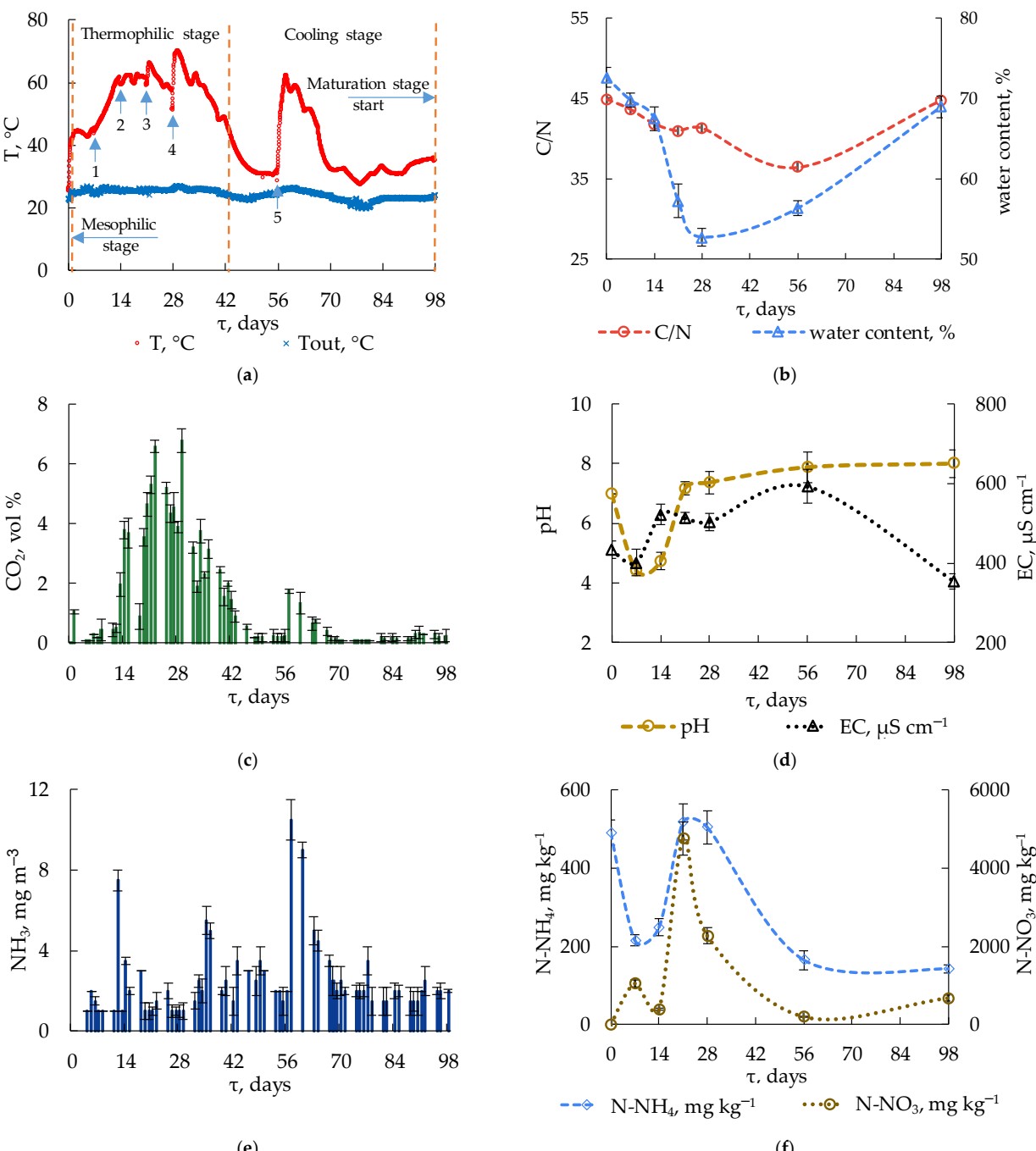

**Figure 2.** Dynamics of physicochemical parameters during composting: (**a**) the temperature regime: T, substrate temperature; Tout, ambient temperature; 1–5—stirring for 2 min; 4 and 5—addition of water; (**b**) mass water content and C/N ratio; (**c**) $CO_2$ production; (**d**) pH and the electrical conductivity (EC); (**e**) ammonia release; (**f**) ammonium and nitrate nitrogen.

The temperature then decreased and reached the same value (44 °C) again only on day 6. During days 0–7 pH decreased to 4.4 due to organic acid formation. Therefore, this period (~4 days) may be a reserve for increasing the process efficiency by using a specific microbial consortium, including degraders of organic acids.

During days 12 to 28, the temperature did not fall below 55 °C; for 12 days it exceeded 60 °C. High temperatures resulted in water evaporation, Figure 2b, and, therefore, a decrease in microbial activity. The stirring of the substrate and water addition on day 28 resulted in an intensification of the OM degradation–$CO_2$ emissions raised to 6.8% (vol/vol) and the temperature raised to the maximum (70 °C) by day 30. The rapid transition to thermophilic conditions and prolonged maintenance of this mode has been described previously [19,44,45].

During days 42 to 56, the temperature decreased to mesophilic values. The addition of water and mixing of the substrate on day 56 resulted in a temperature increase and repeated activation of the microbial activity according to raised $CO_2$ emissions. During days 56 to 63, the temperature increased, peaking at 62 °C. On day 98, the temperature differed from the ambient values by 10–12 °C, and mixing did not result in significant increases in activity and temperature.

High $CO_2$ emissions exhibited a positive correlation with high temperatures during days 13–42 of composting, Figure 2c. While 87% of easily degradable OM was mineralized to $CO_2$ during the first 42 days, only 13% was mineralized during the next 56 days.

During the active period, the $CO_2$ concentration was as high as 5.3–6.8% (vol/vol). Intense $CO_2$ production during the first 20 days has also been previously reported [46,47].

The measured C–$CO_2$ mineralization (rate of microbial respiration) during days 13–42 was, on average, 145 mg $CO_2$ kg$^{-1}$ h$^{-1}$ (per dry mass), with the maximum at 282 mg $CO_2$ kg$^{-1}$ h$^{-1}$. On day 98, the rate of microbial respiration was 17 mg $CO_2$ kg$^{-1}$ h$^{-1}$, which was below the maximal respiration rate ($\leq$200 mg $CO_2$ kg$^{-1}$ h$^{-1}$) proposed for stabilized compost [21]. The differences in the values of microbial respiration may be due to the differences in the applied procedure: direct measurement at dynamic self-heating used in the present work and the method of incubation at constant temperature [21].

The total OM mineralization during the 30-day thermophilic stage was 104.37 g $CO_2$ kg$^{-1}$ (per dry mass). The measured carbon mineralization included C–$CO_2$ resulting from microbial degradation; however, no $CH_4$ was produced in the anaerobic processes.

During the cooling stage (after day 42), the average $CO_2$ concentration was 0.3% (vol/vol). The period from days 57 to 60 was exceptional in this respect, with increasing $CO_2$ concentrations, which reached a local maximum of 1.8% (vol/vol) on day 57.

### 3.1.2. pH and Electrical Conductivity

At the onset of composting, the pH of the substrate was neutral (6.9) due to the application of fresh food and agricultural waste, Figure 2d.

By day 7, the pH decreased significantly (to 4.4). Similar results were obtained by other researchers [48]. The pH value remained low (4.7) until day 14. The low pH resulted from the active production and accumulation of organic acids in the substrate during this period. Because anoxic microniches often occur in the aerobic composted matrix due to limited oxygen inflow, organic acids may be produced by obligate and facultative anaerobes [48]. From day 14 to day 21, the pH increased sharply to 7.2; organic acids were consumed by the microbial community, which resulted in a pH increase [49]. No significant pH changes were observed during days 21–98: the neutral reaction of the substrate very slowly changed to weakly alkaline (pH 8.0 on day 98).

The EC of the initial substrate was 433.5 μS cm$^{-1}$, which was below the previously reported value of 900 μS cm$^{-1}$ for food waste [50]. During the first 7 days of composting, the EC decreased to 401 μS cm$^{-1}$. The EC subsequently increased, reaching 522.3 μS cm$^{-1}$ on day 14. This local maximum was likely caused by the release of mineral salts and ammonium ions due to active OM decomposition, as well as by water loss and increased salt concentration [51].

Apart from inorganic ions, a high concentration of dissolved carbon in the water extract may also affect the conductivity [52]. By day 56 of the experiment, the EC reached its maximum (592.3 $\mu$S cm$^{-1}$). Because the easily available compounds were consumed by this time, an EC decrease began [48]. In the course of further composting, the EC decreased to 353 $\mu$S cm$^{-1}$ on day 98. Ammonia escape and the microbial assimilation of mineral salts were likely responsible for the EC decrease during the cooling stage [51].

### 3.1.3. Dynamics of Nitrogen Compounds

Nitrogen is present in FW mainly as organic nitrogen, which occurs in diverse molecular forms, including proteins, amino acids, nucleic acids, chitins, etc. [1]. Organic nitrogen was mineralized to ammonia ($NH_3$) in the water phase simultaneously with OM degradation. The ammonia emissions during the experiment varied from 1 to 3.5 mg m$^{-3}$ of aerating air, with the exception of days 12, 35, and 57 with the values of 7.5, 5.5, and 10.5 mg m$^{-3}$, respectively, Figure 2e.

Shifting from the mesophilic mode to the thermophilic mode and self-heating of the substrate to 57–60 °C resulted in significant $NH_3$ emissions, with its concentration in waste gas increasing five- to seven-fold. Continued active degradation at high temperatures (65–70 °C) during days 28–32 resulted in a local two-fold increase of $NH_3$ emission on days 35–36.

High temperature plays an important part in this process [12,53]. The ammonium content in the compost decreased from 490 to 215–248 mg kg$^{-1}$ on days 7–14 of composting due to the ammonia release from the substrate, Figure 2f. The formation of ammonia and ammonium ions results from the decomposition of nitrogen-containing compounds, which are abundant at the onset of the process [54]. By day 21, the content of ammonium nitrogen increased to the initial level and then decreased again to 165 mg kg$^{-1}$ on day 56. The ammonium concentration in the substrate did not change significantly afterwards and, in spite of considerable ammonia release on day 57, it was 143 mg kg$^{-1}$ by day 98.

The significant ammonia release on day 57 likely resulted from the death and decomposition of the mesophilic microbiota, which developed during the mesophilic stage (days 42–56) but did not survive the high temperatures during the later stage. The ammonium content by the end of composting was 143 mg kg$^{-1}$, which indicates the stability of the waste after composting [10]. Both ammonium and nitrate are nitrogen species that are available to microorganisms.

A drastic increase in the content of nitrate nitrogen (from 394 to 4745 mg kg$^{-1}$) was observed on days 14–21. Similar to the dynamics of ammonium nitrogen, the amount of nitrate nitrogen decreased with time, reaching 200 mg kg$^{-1}$ on day 56. Nitrate is a nitrogen species available to microorganisms, which likely resulted in its active consumption by the microbial community.

On day 56, the nitrification index (NI), calculated as the N-NH$_4$/N-NO$_3$ ratio, was 0.83 (3 > NI (56) > 0.5), which indicated the stability of the substrate [55]. A further increase in N-NO$_3$ and decrease in N-NH$_4$ during composting resulted in NI = 0.21 on day 98 (0.5 > NI (98)> 0.16), which indicated that the compost was at the maturation stage but did not mature completely [10,55].

### 3.1.4. C/N Ratio and Germination Index

The content of total organic carbon (C, %) decreased during composting from 48.4 to 36.9% by dry matter. Carbon was consumed most extensively (to 37.8%) during the first 56 days. Afterward, as the easily available carbon was consumed, both the microbial activity and the rate of OM decomposition decreased [56].

The Kjeldahl total nitrogen in the substrate did not change significantly during the first 56 days (from 1.08% to 1.04%). During this period, various molecular species of organic nitrogen present in mixed food and agricultural waste was either assimilated by microorganisms or mineralized and lost with released ammonia. The relative N value (%) was maintained due to the concentration effect, as was reported in our earlier work on the

composting of anaerobically processed wastewater sludge [22]. Considerable ammonia emissions during the period from 56 to 98 days resulted in a decrease in the total nitrogen to 0.82%. For this reason, the calculated C/N value was restored during the cooling stage after the decrease during the thermophilic stage (see Figure 2b).

The germination index (GI, %) is an important parameter for assessing compost maturity and toxicity to plants. The GI is known to increase in the course of decomposition of toxic compost components, such as short-chained volatile fatty acids and primarily acetic acid [57]. GI increased during the first 7 days and then decreased to day 28, likely due to the production of excessive ammonium and organic acids. Similar phases of GI inhibition have been previously reported [57]. After 28 days, the GI increased steadily to 107–118%, and the compost ceased to be phytotoxic and reached the standard of compost safety (>100%) [58].

No emission of hydrogen sulfide ($H_2S$) was observed within the studied concentration range, which was likely due to the chosen optimal aeration rate in order to prevent the emission of volatile sulfur compounds [23].

### 3.1.5. Mature Compost

The period of complete compost maturation was ~10 months. The compost sample collected on day 389 met the present criteria, characterized as mature compost [58] and had high agrochemical parameters, Table 3.

**Table 3.** Agrochemical and sanitary microbiological parameters of mature compost.

| Parameters | Units | Value * | Limits |
|---|---|---|---|
| pH | pH units | 7.7 ± 0.1 | 6.0–8.5 [28] |
| EC | μS cm$^{-1}$ | 244 ± 12 | <<4000 [59] |
| Water content | % | 36.68 ± 0.9 | <40 [58] |
| N | % | 1.79 ± 0.20 | 1.6–1.8 [21,51] |
| N-NH$_4$ | mg kg$^{-1}$ | 200 ± 20 | n.d. |
| N-NO$_3$ | mg kg$^{-1}$ | 1203 ± 192 | n.d. |
| NI | | 0.16 | <0.5 [58]; 0.16 [10,55] |
| OM | % | 50.9 ± 0.9 | >35 [58] |
| C | % | 28.3 ± 0.9 | n.d. |
| C/N | | 15.8 | <20 [58]; <25 [28] |
| Phosphorus as P$_2$O$_5$ | % | 1.15 ± 0.05 | n.d. |
| Potassium as K$_2$O | % | 1.25 ± 0.1 | n.d. |
| Humic acid (HA) mass | g kg$^{-1}$ | 52.2 ± 5.2 | HA + FA > 7% [60] |
| Fulvic acid (FA) mass | g kg$^{-1}$ | 200 ± 20 | HA + FA > 7% [60] |
| GI | % | 178 ± 14 | >100 [58] |
| Coliform bacteria | CFU per 1 g | 1 | 1 [30]; ≤1000 CFU g$^{-1}$ [28] |
| Pathogenic micro-organisms | in 25 g | not detected | absent [28,30] |
| Enterococci, index | CFU per 1 g | not detected | <100 [28] |
| Protozoan cysts and helminth larvae and eggs | in 1 g | not detected | absent [28] |
| Pupae and larvae of synanthropic flies | ind. from 0.2 × 0.2 m area | not detected | n.d. |

* Data expressed as mean ± standard deviation (*n* = 3).

Mature compost from mixed food and agricultural waste had a close-to-neutral pH of 7.7. Similar pH dynamics of composted FW from neutral to acidic and then from alkaline to neutral were formerly reported [50]. The water content was 40% (wt/wt), which was in agreement with the requirements for compost quality [58].

Mature compost had an EC of 244 $\mu S\ cm^{-1}$, i.e., considerably lower than the critical threshold (4000 $\mu S\ cm^{-1}$) that is considered required for plant growth [59]. High EC values indicate excessive concentrations of soluble salts and are considered undesirable for plant growth and development [50].

The nitrification index of the final product (NI) was 0.16, which is considered one of the major indicators of complete compost maturity [10,55,58].

The C/N ratio was in agreement with the values for mature compost [58]. The average nitrogen content in the product was within the characteristic values (1.6–1.8%), and the OM (51%) mass share exceeded both these values for the compost from kitchen waste (39–45%) [21,51] and the minimal value for usable compost (35%) [58]. Mature compost from mixed food and agricultural waste was characterized by high levels of phosphorus and potassium and had a stimulating effect on plant growth (GI 178%). The sanitary microbiological parameters were in agreement with accepted standards [28,30].

Humification of stabilized compost occurs during the maturation stage. To meet agricultural requirements, the product should have low toxicity and a high level of humification [58]. Mature compost contains significant amounts of humic and fulvic acids (up to 252 $g\ kg^{-1}$). This may be considered a characteristic feature of compost quality, because, to prevent nitrogen immobilization, the carbon of organic fertilizers should not be easily susceptible to microbial attacks [21].

Thus, co-composting of a mixture of FW and wheat straw (85:15) in a laboratory experimental system (in-vessel composting system) under active aeration for 98 days, until the formation of a stable product, followed by a 10-month maturation in a heap with natural ventilation, resulted in the production of mature compost satisfying the quality criteria. The high initial C/N ratio of 45, based on conventional C determination, did not result in increased nitrogen immobilization and decreased the levels of the forms of the elements of plant nutrition. The mesophilic and thermophilic stages of composting were carried out for 42 days, and the temperature maximum of 70 °C was reached.

The active mineralization of organic matter (by $CO_2$ release) occurred for 30 days during the thermophilic stage and was, on average, 145 mg $CO_2\ kg^{-1}\ h^{-1}$. The shift from the mesophilic to the thermophilic mode resulted in a five- to seven-fold increase in ammonium emissions during a 24 h period. The period from days 56 to 98 was sufficient for the stabilization of organic compounds and for the loss of toxicity to plants. Subsequent incubation of the product in a simple compost heap resulted in lignocellulose decomposition and of the formation of humic compounds in mature compost. Mature compost had a N–P–K ratio (nitrogen (N), phosphorus (P), potassium (K)) of 1.79–1.15–1.25; and its total content of humic and fulvic acids was 252 $g\ kg^{-1}$.

*3.2. Total Microbial Number at Different Stages of Composting*

The numbers of cultured aerobic heterotrophic microorganisms were monitored during composting using CFU counts on solid medium. The number of mesophilic microorganisms ($3 \times 10^6$ CFU $g^{-1}$) was initially higher than the number of thermophilic microorganisms ($4 \times 10^5$ CFU $g^{-1}$), Figure 3a.

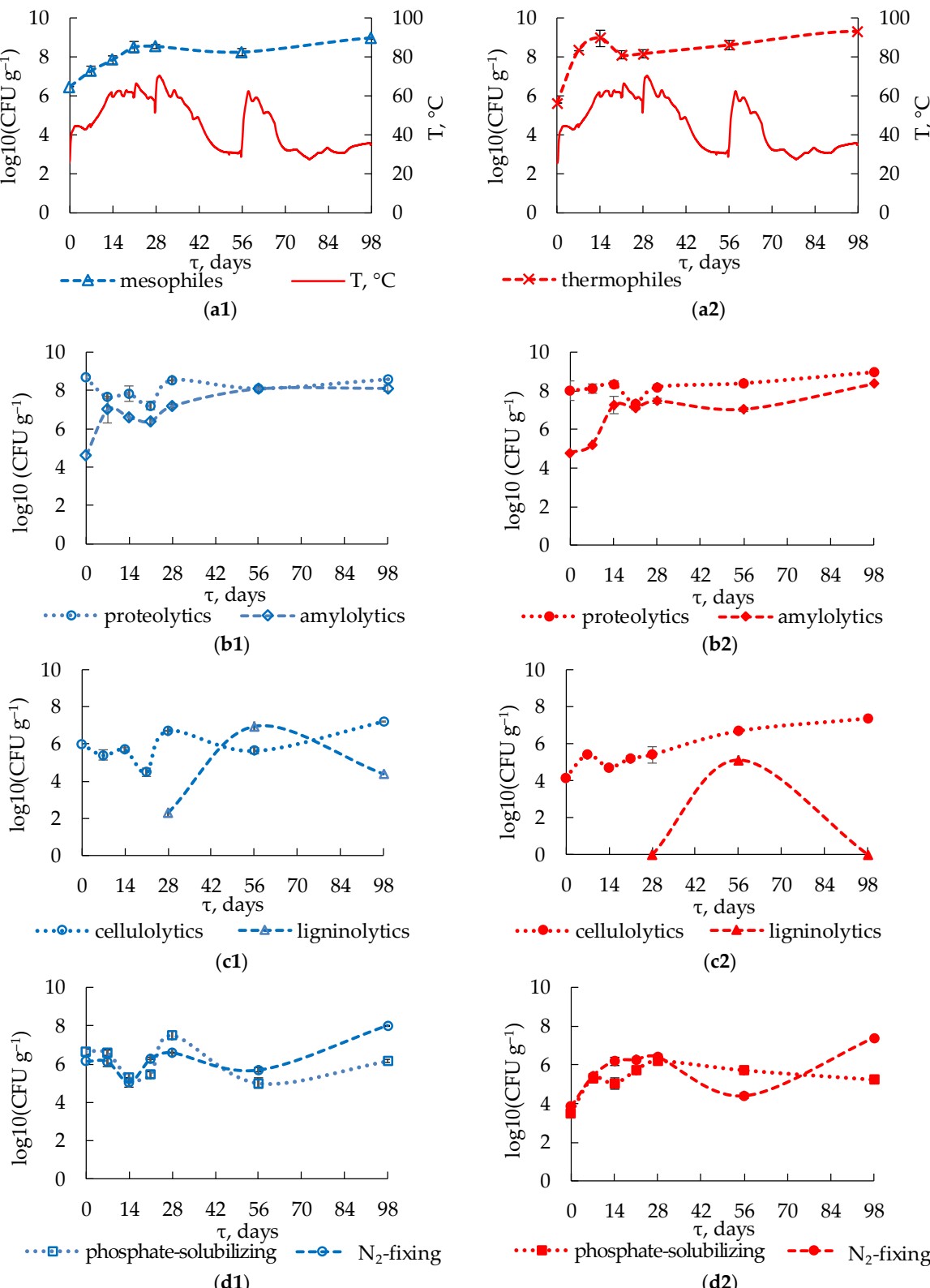

**Figure 3.** Dynamics of cultured microorganisms during composting: (**a**) total microbial number in mesophilic and thermophilic conditions: T, substrate temperature; (**b**) proteolytic and amylolytic microorganisms; (**c**) cellulolytic and ligninolytic microorganisms; (**d**) phosphate-solubilizing and nitrogen-fixing microorganisms; 1, mesophilic microorganisms; 2, thermophilic microorganisms.

The abundance of mesophilic microorganisms increased throughout the composting process, with an especially active increase during the first 21 days to $3 \times 10^8$ CFU $g^{-1}$. During days 21 to 98, the number of mesophilic microorganisms varied within the order of magnitude of $10^8$ CFU $g^{-1}$, reaching $6 \times 10^8$ CFU $g^{-1}$ by the end of the cooling phase. Many mesophilic microorganisms likely survived the thermophilic conditions due to their ability either to grow within a broad temperature range or to form a persistent form to survive under unfavorable conditions. A similar dynamic was previously observed for mesophilic composting [61], while other researchers reported an abundance of mesophilic heterotrophic microorganisms that peaked on day 21 and decreased afterwards [62].

The number of thermophilic microorganisms increased in 14 days by three orders of magnitude and reached the first maximum of $9 \times 10^8$ CFU $g^{-1}$. Significant growth of thermophiles was due to the temperature increasing from 40 to 60 °C during this period. Maintaining the temperature close to 60 °C from day 14 on led to intense water evaporation (see Figure 2b) and a subsequent decrease in microbial activity and number. On days 21–28 of composting the number of microorganisms, growing at 50 °C decreased ten-fold.

Afterward, however, the number of thermophiles increased again, reaching the second maximum of $2 \times 10^9$ CFU $g^{-1}$ on day 98, in spite of the mesophilic conditions at the end of composting. The numbers of mesophilic and thermophilic microorganisms at the early maturation phase (day 98) agreed with the values reported in other works using the same enumeration procedure [61,63], as well as with the qPCR results [17]. Other researchers reported the highest number of aerobic heterotrophic microorganisms (~$10^9$ CFU $g^{-1}$) on days 40–60 of composting (thermophilic conditions) [64].

Thus, according to the results of culture-dependent analysis, the total microbial abundance increased with composting time and was significantly higher on day 98 rather than on day 0.

### 3.3. Abundance of Various Physiological Groups of Cultured Microorganisms during Composting

Classical microbiological methods were used to investigate seven metabolic groups of microorganisms under mesophilic (28 °C) and thermophilic (50 °C) conditions. Proteolytic microorganisms were abundant in the compost material at all stages, with an average number of $10^8$ CFU $g^{-1}$, Figure 3b.

The lowest number of proteolytic microorganisms ($10^7$ CFU $g^{-1}$) was observed on day 21. The abundance of mesophilic and thermophilic microorganisms changed in a similar manner. Previous research also reported high numbers of proteolytic microorganisms (~$10^7$ CFU $g^{-1}$) throughout the composting process [62].

The numbers of mesophilic amylolytics increased from $4 \times 10^4$ to $1 \times 10^7$ CFU $g^{-1}$ during days 0–7, (see Figure 3b). Active growth of thermophilic amylolytics occurred from day 7 on, when the compost material was heated to 40 °C. During days 14–28 (under thermophilic conditions), the number of mesophilic amylolytic microorganisms decreased to $2.4 \times 10^6$ CFU $g^{-1}$, while the number of thermophiles increased to the local maximum of $3.0 \times 10^7$ CFU $g^{-1}$.

Other researchers [63,64] also reported active growth of this group during the first two weeks of composting. The highest abundance of both groups was observed during the maturation stage beginning (98 days), likely due to the formation of the most favorable conditions for their growth. A similar pattern was formerly noted [61]. Amylolytic and proteolytic bacteria are known to be involved in the synthesis of humic compounds [65].

The number of thermophilic cellulolytics increased gradually from $1.4 \times 10^4$ to $2.3 \times 10^7$ CFU $g^{-1}$ throughout almost the whole period of composting, Figure 3c.

Similar dynamics were formerly reported [61]. A local maximum of $2.5 \times 10^5$ CFU $g^{-1}$ on day 7, followed by a decrease in microbial number to $5.3 \times 10^4$ CFU $g^{-1}$ by day 14, was an exception to this pattern. Before day 21, the number of mesophilic microorganisms did not exceed the initial value of $1.0 \times 10^6$ CFU $g^{-1}$, while, on day 28, their number increased sharply from $3.0 \times 10^4$ to $5.2 \times 10^6$ CFU $g^{-1}$. These results agree with the data reported earlier [66]. Their number decreased to $4.3 \times 10^5$ CFU $g^{-1}$ on day 56 and then increased

again to $1.6 \times 10^7$ CFU g$^{-1}$ on day 98, likely due to the more favorable growth conditions caused by decreased amount of easily degradable substrates and decreased abundance of the relevant micro-organisms, which were predominant in the substrate during the previous stages.

As ligninolytic microorganisms develop in the compost close to its maturation stage [25], the analysis of this group was commenced on day 28 of the process. Fungi represented ligninolytic communities obtained at different stages. The number of mesophilic ligninolytics on day 28 was $2.0 \times 10^2$ CFU g$^{-1}$, (see Figure 3c). When mesophilic conditions were established, their number increased to $9.0 \times 10^6$ CFU g$^{-1}$ on day 56 and then decreased to $2.4 \times 10^4$ CFU g$^{-1}$ on day 98. Thermophilic ligninolytic fungi were detected only on day 56 ($1.3 \times 10^5$ CFU g$^{-1}$). Lignin degradation was likely more active under mesophilic conditions, which were preferable for the growth of mycelial forms.

Apart from the microorganisms utilizing various carbon substrates as energy sources, microbial groups involved in nitrogen and phosphorus metabolism were studied: phosphate-solubilizing, nitrogen-fixing, and nitrifying microorganisms. The initial abundance of mesophilic nitrogen-fixers in the substrate was $1.4 \times 10^6$ CFU g$^{-1}$, Figure 3d.

Their number decreased on days 7–14 under thermophilic conditions and then increased to almost the initial values by day 28, in spite of the high temperature of the system. Their highest number was found on day 98 ($9.9 \times 10^7$ CFU g$^{-1}$). Other researchers observed another growth pattern for mesophilic nitrogen fixers [61,66]: their number decreased gradually with the composting time.

The number of thermophilic nitrogen-fixing microorganisms increased under thermophilic conditions from $6.9 \times 10^3$ CFU g$^{-1}$ on day 0 to $2.5 \times 10^6$ CFU g$^{-1}$ during the first 28 days of composting. Their number then decreased to $2.5 \times 10^4$ CFU g$^{-1}$ on day 56, because the compost temperature on days 45–56 was below 40 °C. Similar to mesophiles, the highest number of thermophilic nitrogen fixers was observed on day 98. Nitrogen-fixing microorganisms contributed to nitrogen accumulation during maturation and, therefore, to increase Kjeldahl total nitrogen in mature compost (see Table 3).

The number of mesophilic phosphate-solubilizers in the original substrate was $4.5 \times 10^6$ CFU g$^{-1}$ (see Figure 3d). On days 14–21, some members of this group died due to the emergence of thermophilic conditions stressful to these organisms. A similar dynamic of the abundance of phosphate-solubilizing microorganisms growing at 30 °C was formerly shown [67,68]; however, on day 28, while the temperature was still high, the abundance of mesophiles reached its maximum ($3.0 \times 10^7$ CFU g$^{-1}$), likely due to active growth of thermotolerant forms. The highest numbers of mesophilic phosphate solubilizers were reported to occur on days 4–7 [67,68] and on day 14 [69] in previous studies. On days 56 and 98, the numbers of the mesophilic members of this group were $1.0 \times 10^5$ and $1.4 \times 10^6$ CFU g$^{-1}$, respectively, i.e., lower than on day 0.

The thermophilic phosphate solubilizers increased their number from $3.0 \times 10^3$ CFU g$^{-1}$ to $1.6 \times 10^6$ CFU g$^{-1}$ by day 28. Former research reported the highest number of thermotolerant microorganisms on day 14 [69]. The onset of mesophilic conditions resulted in a slight decrease in their abundance to $5.2 \times 10^5$ CFU g$^{-1}$ on day 56 and to $1.7 \times 10^5$ CFU g$^{-1}$ on day 98, which was typical of thermophiles. Other researchers also observed a decreased abundance of phosphate-solubilizing microorganisms by the maturation phase, which was caused by consumption of organic substrates for microbial growth [69].

The nitrification process consists of two subsequent stages-the oxidation of ammonium to nitrite (I) and the oxidation of nitrite to nitrate (II). The abundance of stage I and stage II nitrifying micro-organisms during composting was assessed by the consumption of ammonium and nitrite, respectively. The percentage of consumption calculated was based on the maximum and minimum content of the substrate (ammonium or nitrite) in the medium during the cultivation period (42 days). Under mesophilic conditions, a decrease in ammonium concentration in the medium was the most pronounced in the samples

collected on days 21 and 28 (53% and 59% ammonium consumed), as well as at the onset of the experiment (32%) and on days 7 and 14 (28% each), Figure 4a.

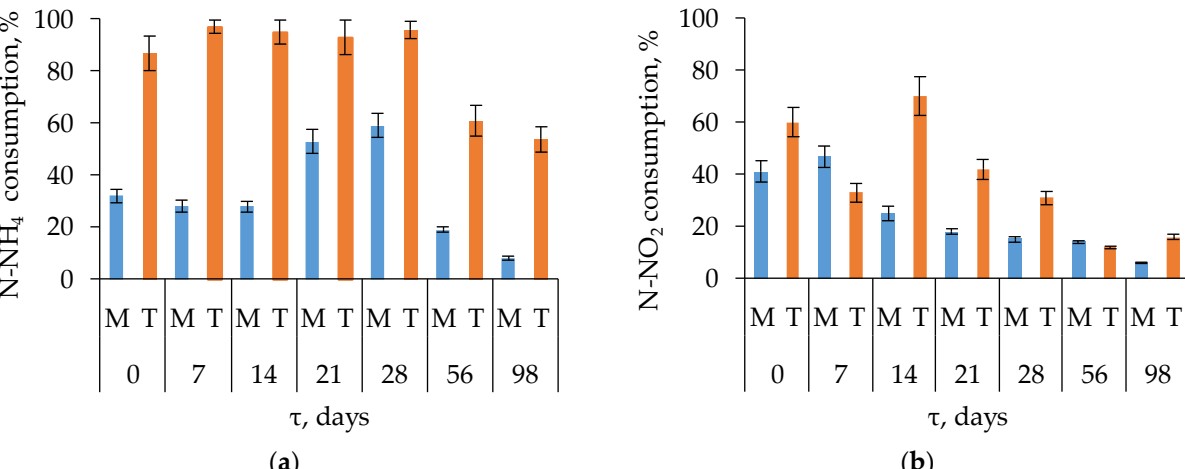

**Figure 4.** Dynamics of nitrifying activity: (**a**) stage I nitrifying microorganisms according to the ammonium consumption; (**b**) stage II nitrifying microorganisms according to the nitrite consumption isolated at the onset of composting (0) and on days 7, 14, 21, 28, 56, and 98 of composting; M, mesophilic microorganisms; T, thermophilic microorganisms.

Although mesophilic conditions were established on days 56 and 98, the activity of stage I mesophilic nitrifiers was low, with consumption of only 19% and 8% ammonium, respectively. This was likely the result of the low ammonium content in the substrate (see Figure 2f). Thermophilic stage I nitrifiers were most active in the samples collected at the onset of composting and on days 7, 14, 21, and 28, with 88–98% ammonium consumed, (see Figure 4a).

In general, thermophiles oxidized ammonium more quickly and efficiently than mesophiles. The highest activity of ammonium-oxidizing prokaryotes was observed during the active phase of composting, during the first 28 days, due to significant concentrations of ammonium ions resulting from the decomposition of nitrogen-containing organic compounds. Similar conclusions were made in former studies [61]. Significant decrease of ammonium (see Figure 2f) due to high nitrifying activity was observed on days 28–56.

Stage II nitrifiers cultivated under mesophilic conditions were especially active at the beginning of composting and on day 7, with nitrite consumption of 41% and 47%, respectively, Figure 4b.

Nitrite consumption in the other samples varied from 6% to 25%, with the lowest activity on day 98 of composting. Under thermophilic conditions, nitrite was rapidly consumed by stage II nitrifiers from the samples collected at the onset of composting and on days 14 and 21, (see Figure 4b). For these samples, the nitrite consumption was 60%, 70%, and 42%. The present work revealed the highest activity of nitrifiers during the active phase of composting (the first 28 days), when active OM decomposition and ammonium release from organic compounds occurred.

The stable high temperature (60–65 °C), production of high amounts of ammonium, and close-to-neutral pH during days 14–28 provided favorable conditions for the development of thermophilic nitrifying microorganisms, which resulted in a decrease in the content of ammonium and a drastic increase in the content of nitrate nitrogen (see Figure 2f). The data on nitrifying activity obtained in the present work agree with a previous study [66]. In some work, the highest nitrification rate was observed during the maturation phase [70–72].

No strong correlation between physicochemical conditions and the activity of cultured microbial physiological groups was found. Hence, physiological groups adjusted to

changing ambient conditions. When the environmental conditions changed, the composition of physiological groups also probably changed, and microorganisms for which these conditions were optimal developed rapidly. Thus, the activity of the physiological group did not decrease. This statement characterizes the compost microbial community as stable. According to the Pearson correlation coefficients, proteolytics, amylolytics, and cellulolytics co-developed in the community. The growth of proteolytics, amylolytics, and cellulolytics was also associated with the growth of nitrogen fixers in the microbiota. The growth of proteolytics and cellulolytics negatively correlated with the growth of ligninolytics. The same groups also negatively correlated with the activity of nitrifiers in compost. The data is represented in Table S1.

All in all, proteolytic microorganisms were numerous at all stages of composting. The active growth of amylolytic microorganisms was observed during the first 14 days of composting. The growth of phosphate solubilizers and the activity of nitrifiers were generally highest during the first month of composting. Cellulolytic microorganisms began to grow in the thermophilic period and reached a maximum number at the beginning of maturation. Researchers [64] have observed a significant increase in amylolitcs, cellulolytics, and proteolytics during the thermophilic stage.

The most significant increase in microbial numbers for ligninolytic and nitrogen-fixing microorganisms was shown during the cooling stage and at the beginning of maturation. As a result of the activity of microbial groups with diverse metabolic possibilities, by day 98, stable compost was formed, from which a considerable amount of easily available OM was removed.

### 3.4. Composition and Biodiversity of the Microbial Community

#### 3.4.1. Fungal Community

The composition of the fungal community was analyzed according to the number of ITS copies. During composting, at least 92% of the fungal community belonged to the phylum *Ascomycota*; in mature compost, the abundance of *Mortierellomycota* and *Basidiomycota* increased to 2% and 1%, respectively. At different stages, members of the genera *Aspergillus*, *Penicillium*, *Byssochlamys*, *Thermomyces*, and *Microascus* were predominant in the material, Figure 5.

Unfortunately, NGS profiling of the fungal community in the original material (day 0) was not successful. This was caused by the prevalence of nondecomposed biomass of *Brassica oleracea*, which contains ITS with similar sequences at primer annealing sites. At latter stages of composting, *B. oleracea* was decomposed, making it possible to analyze the composition of the fungal community.

*Byssochlamys* predominated in the substrate before composting (80%) and on day 56 (96%), while its share on day 98 was only 2% (Figure 5). This genus forms heat-resistant spores, surviving at 85 °C, grows at low oxygen concentrations, and is involved in the decomposition of plant material [73]. Certain strains produce mycotoxins and immunosuppressing agents [74]. *Byssochlamys* survived high-temperature composting conditions due to persistent spores and then recolonized the substrate on day 56, when mesophilic conditions were restored. The decreased abundance of this fungus in the community by day 98 indicated the safety of the prepared compost.

During the thermophilic phase of composting, members of the genera *Aspergillus* and *Penicillium* contributed most significantly to the OM decomposition. The abundance of these genera in the compost remained almost stable during days 7–28, with *Aspergillus* and *Penicillium* responsible for 36–41% and 22–26% of the fungal biota, respectively; their relative abundance on day 98 was 35% and 6%, respectively. *Aspergillus* and *Penicillium* were often found to be the dominant genera of compost and soil fungal communities [75]. On day 7, the compost contained ~2% of *Candida*, while, on day 28, the genus *Cyberlindera* constituted ~1%.

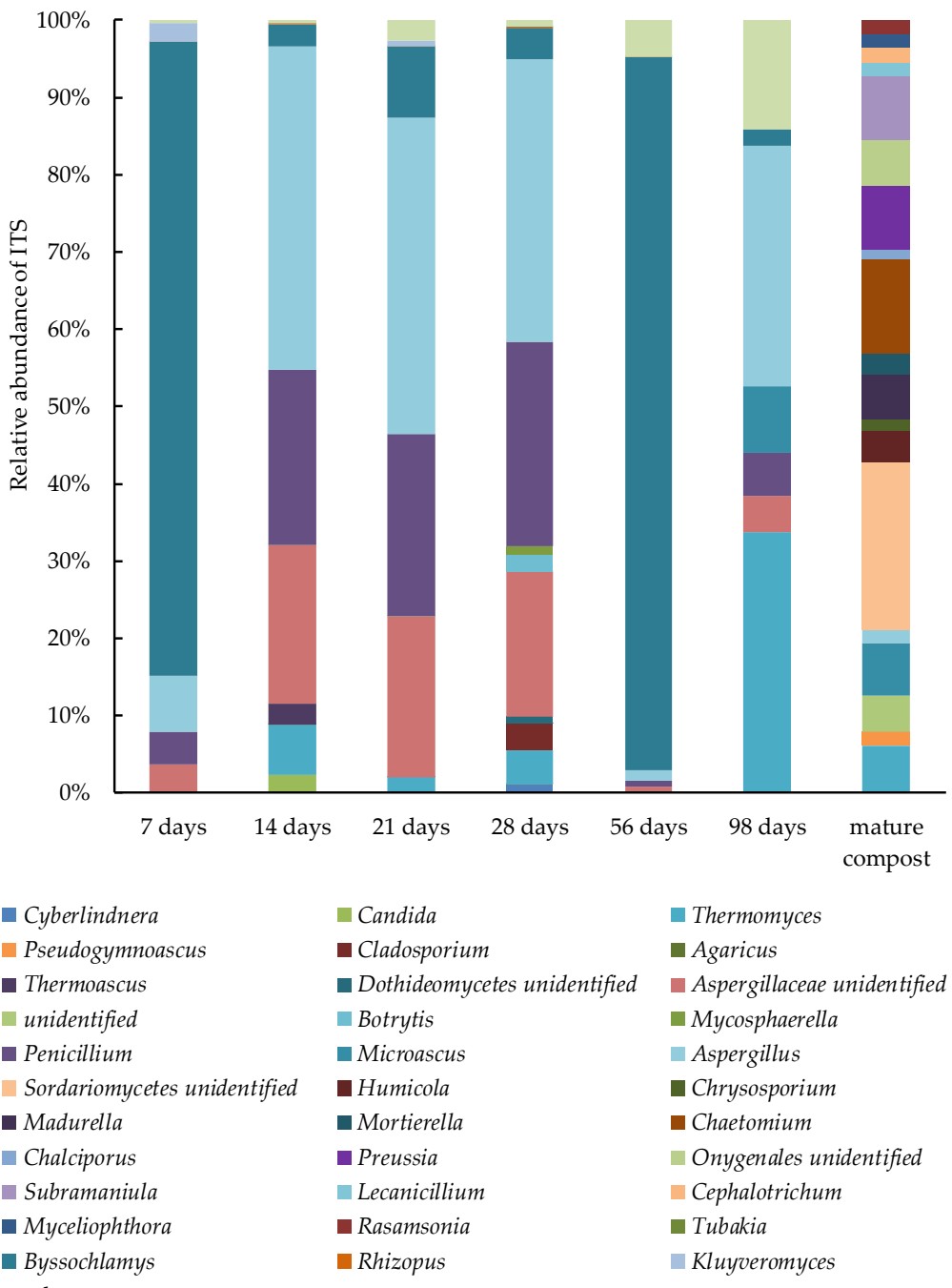

**Figure 5.** Genus-level composition of the fungal community (genera or higher taxa of fungi that constituted more than 1% of the fungal community are represented; the rest are grouped under the name "other").

These were likely the remnants of the diverse yeasts that are typically predominant at the initial stage of composting under mesophilic conditions and the high content of easily degradable organic compounds [17]. While *Thermomyces* constituted 6% on day 7 and 4% on day 21 of composting, it was predominant in the community on day 98 (37%). It was the dominant fungus at the end of large-scale agrowaste composting and produced various hydrolytic enzymes [19]. On day 98, members of the genus *Microascus* were found in the compost community (9%).

According to other authors, *Microascales* was the predominant group during compost maturation [17,76]. According to the results of sequencing mature compost samples,

the following taxa were predominant in the mycobiota: class *Sordariomycetes* (18%) and genera *Chaetomium* (10%), *Preussia* (7%), *Subramaniula* (7%), *Madurella* (5%), and *Humicola* (4%), which were not previously revealed in compost samples in this work, as well as *Thermomyces* (5%) and *Microascus* (6%), which were predominant at the previous stages.

*Chaetomium* and the related genera *Subramaniula* and *Humicola* are soil micro-organisms specializing in the degradation of cellulose and lignin; these genera include thermophilic and thermotolerant members [77]. The decomposition of poorly degradable polymers by these fungi likely resulted in the formation of humus components and the accumulation of humic substances in mature compost (see Table 3). The abundance of *Sordariomycetes* also increased at the completion of the process according to Galitskaya et al. (2017) [17].

Therefore, the predominant members of the fungal community were *Byssochlamys* at the start of the thermophilic stage and *Aspergillus* and *Penicillium* during the main part of the thermophilic stage. *Byssochlamys* was dominant also during the cooling stage. During maturation, *Thermomyces* and *Microascus* prevailed. In mature compost, there were more dominant groups represented—*Sordariomycetes*, *Chaetomium*, *Preussia*, *Subramaniula*, and others.

### 3.4.2. Prokaryotic Community

In the prokaryotic community, the phyla *Firmicutes*, *Actinobacteria*, and *Proteobacteria* were abundant in the composting process. There were few representatives of the archaeal domain. They made up much less than 1% of the total prokaryotic community. The reason for their low numbers was that archaea are usually oligotrophic and develop more slowly than bacteria [78]. Archaea grew poorly in rapidly changing composting conditions. On days 56–98, members of the phyla *Chloroflexi* and *Gemmatinomonadetes* constituted 1–3%, and, on day 98, *Bacteroidetes* were detected (7%).

The phyla found on day 98 were also present in mature compost. *Firmicutes* predominated in the community of the compost, especially at the early stage. The abundance of members of *Actinobacteria* increased gradually during composting (including the high-temperature stage). A similar picture of the prokaryotic microbiota at the phylum level was previously reported [19]. While *Proteobacteria* constituted a minor component of the microbiota during the active phase of composting, they became more numerous during cooling and were responsible for 37% of the community on day 98. In mature compost, however, their relative abundance decreased again.

Prior to the onset of composting, genera of the order *Lactobacillales* were predominant in the substrate: *Leuconostoc* (53%), *Lactococcus* (9%), *Weisella* (3%), and *Vagococcus* (3%), Figure 6.

These bacteria consumed easily available organic compounds (primarily sugars) by fermentation and released organic acids [16,17], that led to a lower pH of the substrate (4.4 on day 7-see Figure 2d) and a slower microbial degradation according to temperature and $CO_2$ emissions (see Figure 2a,c). On day 7, lactic acid bacteria still constituted a significant part of the prokaryotic community: *Weisella* (31%), *Limosilactobacillus* (24%), *Leuconostoc* (5%), while the relative abundance of *Bacillus* increased to 25%, as compared to 2.5% at the onset of composting. On day 14, members of the order *Bacillales* constituted almost 100% of the community: *Bacillus* spp. (35%), *Ureibacillus* spp. (12%), *Aeribacillus* spp. (10%), *Thermobacillus* spp. (4%), *Anoxybacillus* spp. (3%), *Caldibacillus* spp. (3%), as well as uncultured *Bacillaceae* (32%). This order prevailed in the prokaryotic community under thermophilic conditions.

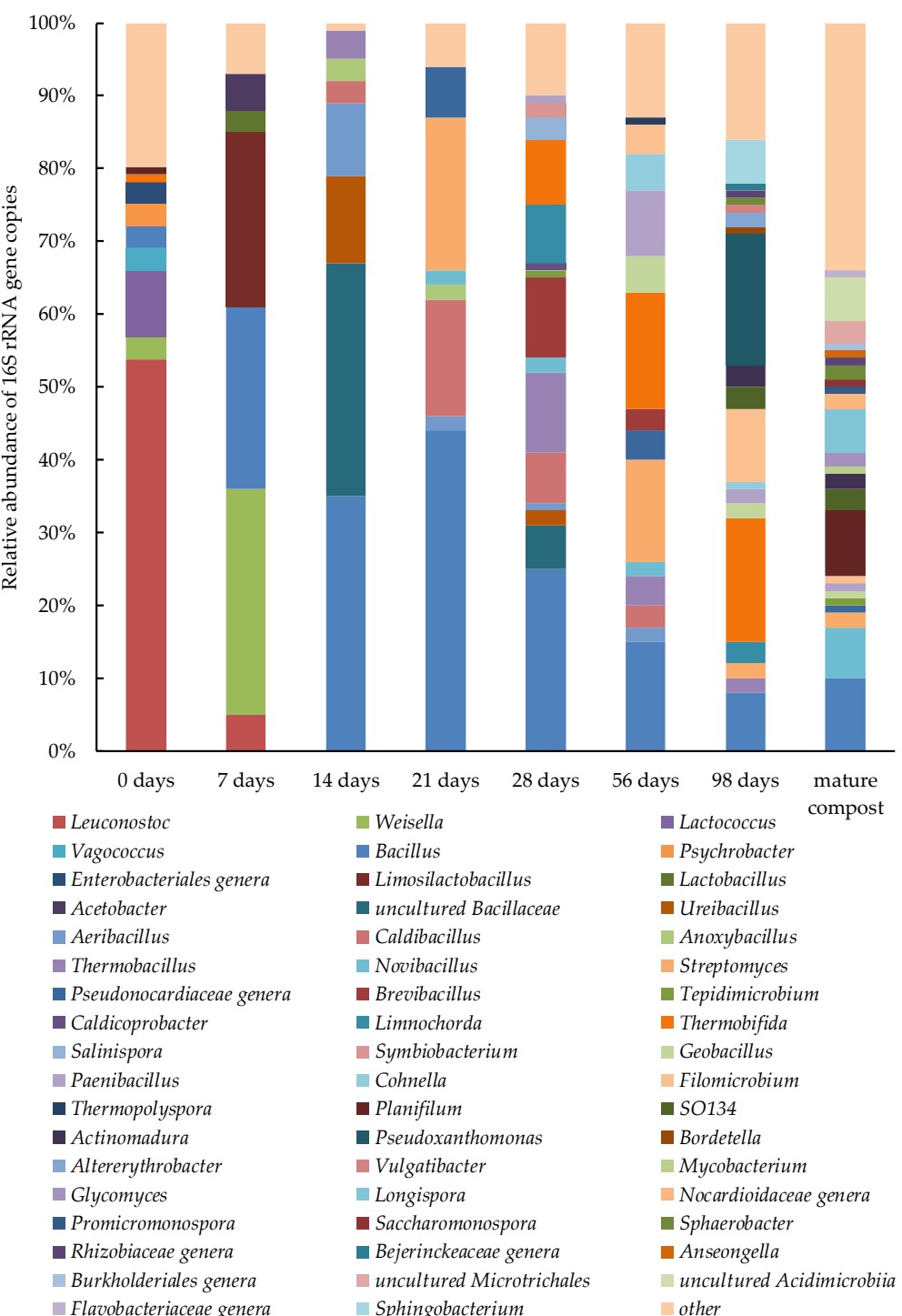

**Figure 6.** Genus-level composition of the prokaryotic community (genera or higher prokaryotic taxa that constituted more than 1% of the prokaryotic community are represented; the rest are grouped under the name "other").

On day 21, bacteria of the genera *Bacillus* (44%) and *Caldibacillus* (16%) still predominated in the compost. Members of the genus *Bacillus* are among the most common micro-organisms colonizing the composted material, as was shown by both classical microbiological and molecular studies [3,16,19,79–81]. *Streptomyces* (21%) and members of the family *Pseudonocardiaceae* (7%) from *Actinobacteria* phylum also grew actively in the compost on day 21.

The genus *Streptomyces* is an important agent of composting for several reasons. Apart from ability of the bacteria to solubilize lignin and cellulose [82,83], they also produce antimicrobial compounds, which suppresses development of human and plant pathogens. *Streptomyces*, as well as *Bacillus*, may be used in biocontrol of plant pathogens in soils [82]. On day 28, the genera *Bacillus* (25%), *Caldibacillus* (7%), *Thermobacillus* (11%), *Brevibacillus* (11%), *Limnochorda* (8%), *Thermobifida* (9%), *Salinispora* (3%), and *Symbiobacterium* (2%) predominated in the community.

Earlier research observed an abundance of members of the cellulolytic genus *Thermobacillus,* which increased during the thermophilic stage of composting [19]. The presence of facultative anaerobes of the genera *Limnochorda* [84] and *Symbiobacterium* indicated the occurrence of microaerobic or anaerobic niches in the compost. It likely resulted from active oxygen consumption by the microbial community during intense biodegradation on day 28 characterized by high temperature and high level of carbon dioxide emission (see Figure 2a,c). On day 56, while the abundance of such genera as *Bacillus*, *Caldibacillus*, etc., which predominated during the thermophilic stage, decreased, new micro-organisms were detected: *Filomicrobium* (4%), *Paenibacillus* (9%), *Cohnella* (5%), and *Geobacillus* (5%).

The abundance of actinomycetes *Thermobifida* (16%) and *Streptomyces* (14%) in the microbiota also increased. According to their metabolic abilities, members of the genus *Paenibacillus* contributed to the degradation of biopolymers and to the increase of available nitrogen and phosphorus; these are among their properties in stimulating plant growth [85]. *Thermobifida* is a widespread compost micro-organism that actively participates in lignocellulose decomposition under both thermophilic and mesophilic conditions [19,86].

The genus *Thermobifida* remained as one of the dominant ones (17%) on day 98. On day 98, the genera *Pseudoxanthomonas* (18%), *Sphingobacterium* (6%), and *Altererythrobacter* (2%), families *Rhizobiaceae* (1%) and *Bejerinckeaceae* (1%) were revealed. The share of *Filomicrobium* increased to 10%, while the share of *Bacillus* decreased still further (to 8%).

The genus *Pseudoxanthomonas* likely contributed significantly to the decomposition of the lignocellulose present in the compost [87]. According to other authors, the share of this genus during the mesophilic and thermophilic stages of composting was 3–4%, while it was hardly detected during the maturation stage [88]; it predominated on days 3 and 13 of composting [89]. Mesophilic bacteria of the genus *Sphingobacterium* are able to utilize sugars with the production of acids; this genus was revealed at late stages of composting in a former study [90]. The 16S rRNA gene sequences of *Sphingobacterium* and *Pseudoxanthomonas* were found to be numerous in the compost of anaerobically digested wastewater sludge in our earlier work [22].

The species predominant in mature compost belonged to the genera *Bacillus* (10%), *Novibacillus* (7%), and *Planifilum* (9%) of the phylum *Firmicutes*; *Longispora* (6%), *Glycomyces* (2%), *Streptomyces* (2%), members of the class *Acidimicrobiia* (9%) belonging to *Actinobacteria*. The micro-organism SO134 (3%) of the phylum *Gemmatinomonadetes*, and *Sphaerobacter* (2%) of the phylum *Chloroflexi*. *Planifilum* is involved in lignocellulose decomposition [91]. This genus was one of the dominants during the cooling phase (11.9%) [92]. Nitrogen-fixing micro-organisms were found in the genus *Planifilum* and family *Bacillaceae* [92].

Members of the family *Bacillaceae* and of the genus *Streptomyces* may be involved in phosphate solubilization [93]. *Longispora* strains are capable of degrading proteins and starch [94]. As *Glycomyces* was revealed in the rhizosphere and in various plant parts as an endophyte [95,96], this organism may be expected to have a positive effect on plant growth. Members of the genus *Sphaerobacter* produce various proteases and cellulases [97,98]. Hydrolytic bacteria (*Longispora, Planifilum, Sphaerobacter,* and *Bacillus*) involved in the degradation of lignin, cellulose, starch, and proteins play a key role in humus formation [99] and its accumulation in mature compost (see Table 3).

Consequently, lactic acid bacteria (*Leuconostoc, Weisella, Lactococcus,* and *Limosilactobacillus*) formed the main part of the community during the mesophilic and heating stages. From the thermophilic stage and until the formation of mature compost, *Bacillus* constituted a significant part of the community. The genera *Ureibacillus*, *Aeribacillus*,

*Thermobacillus,* and *Caldibacillus* also dominated during the thermophilic stage. During cooling, actinomycetes (*Thermobifida, Streptomyces*) predominated in the community, as well as *Paenibacillus, Geobacillus,* and *Cohnella.* The main components of the community during the maturation stage were *Pseudoxanthomonas, Thermobifida, Filomicrobium,* and *Sphingobacterium. Planifilum, Novibacillus,* and *Longispora* were the main members of mature compost community.

To summarize the results obtained, we have included the main functional groups, microbial genera, and physicochemical parameters at each stage of composting in Table 4. Microbial genera that constituted more than 5% of the community were used in the table.

**Table 4.** The main physiological groups, prokaryotic and fungal genera, and physicochemical properties of the four composting stages and mature compost.

| Compost Stage | Time, Days | Physiological Groups | High Taxon | Genus | % | T, °C | pH | EC, µS cm⁻¹ | NI | $CO_2$, vol% | $NH_3$, mg m⁻³ | N-$NH_4$, mg kg⁻¹ | N-$NO_3$, mg kg⁻¹ | C/N |
|---|---|---|---|---|---|---|---|---|---|---|---|---|---|---|
| Mesophilic | 0–1 | amylolitics M, cellulolytics T, $N_2$-fixers M, T, P-solubilizers M, T (CFU g⁻¹ number increase on 0–7 days) | Bacteria | *Leuconostoc* / *Lactococcus* | 53 / 9 | 25.4 ± 0.1 | 6.9 ± 0.4 | 434 ± 21 | >>3 | 0.0 ± 0.0 | 0.0 ± 0.0 | 490 ± 32 | 0.2 ± 0.1 | 44.8 |
| | | | Fungi | n.d. | | | | | | | | | | |
| | | | Archaea | - | 0 | | | | | | | | | |
| Thermophilic | 1–43 | amylolitics T, cellulolytics M, $N_2$-fixers T, P-solubilizers T, nitrifiers M, T (highest $NH_4^+$, $NO_2^-$ consumption) proteolytics M (CFU g⁻¹ number increase on 7–28 days) | Bacteria | *Bacillus* / *Weisella\** / *Limosilactobacillus\** / *Caldibacillus* / *Aeribacillus* / *Ureibacillus* / *Thermobacillus* | 32 / 31 / 24 / 9 / 5 / 5 / 5 | 56.7 ± 6.2 | 5.9 ± 0.3 | 485 ± 25 | 0.18 | 3.3 ± 0.3 | 1.6 ± 0.4 | 371 ± 27 | 2113 ± 224 | 41.3 |
| | | | Fungi | *Aspergillus* / *Byssochlamys* / *Penicillium* / *Thermomyces* | 32 / 24 / 19 / 5 | | | | | | | | | |
| | | | Archaea | CG1-02-32-21, *Ferroplasmaceae* | 0.02 | | | | | | | | | |
| Cooling | 43–98 | amylolitics M, proteolytics T, liginolytics M, T, cellulolytics T (CFU g⁻¹ number increase on 28–56 days) | Bacteria | *Thermobifida* / *Bacillus* / *Streptomyces* / *Paenibacillus* / *Geobacillus* / *Cohnella* | 16 / 15 / 14 / 9 / 5 / 5 | 31.7 ± 5.4 | 7.9 ± 0.4 | 592 ± 42 | 0.82 | 0.3 ± 0.2 | 2.0 ± 0.1 | 165 ± 24 | 200 ± 14 | 36.4 |
| | | | Fungi | *Byssochlamys* | 96 | | | | | | | | | |
| | | | Archaea | - | 0 | | | | | | | | | |
| Maturation | 98–389 | cellulolytics M, T, amylolitics T, proteolytics M, T, $N_2$-fixers M, T, P-solubilizers M (CFU g⁻¹ number increase on 56–98 days) | Bacteria | *Pseudoxanthomonas* / *Thermobifida* / *Filomicrobium* / *Bacillus* / *Sphingobacterium* | 18 / 17 / 10 / 8 / 6 | 35.1 ± 6.1 | 8.0 ± 0.4 | 353 ± 19 | 0.21 | 0.3 ± 0.2 | 2.0 ± 0.2 | 143 ± 11 | 680 ± 74 | 44.7 |
| | | | Fungi | *Thermomyces* / *Aspergillus* / *Microascus* / *Penicillum* | 37 / 35 / 10 / 6 | | | | | | | | | |
| | | | Archaea | *Methanothermobacter, Methanobacterium, Methanoregula* | 0.03 | | | | | | | | | |
| Mature compost | 389 | n.d. | Bacteria | *Bacillus* / *Planifilum* / *Novibacillus* / *Longispora* | 10 / 9 / 7 / 6 | 19.4 ± 0.1 | 7.7 ± 0.1 | 244 ± 12 | 0.16 | 0.0 ± 0.0 | 0.0 ± 0.0 | 200 ± 20 | 1203 ± 192 | 15.8 |
| | | | Fungi | *Chaetomium* / *Preussia* / *Subramaniula* / *Microascus* / *Thermomyces* | 10 / 7 / 7 / 6 / 5 | | | | | | | | | |
| | | | Archaea | *Nitrososphaeraceae, Hadarhaeales, Methanobrevibacterium* | 0.04 | | | | | | | | | |

\*-detected at the thermophilic stage only on day 7. Physiological groups include aerobic mesophilic (M) and thermophilic (T) microorganisms during composting. Data expressed as mean ± standard deviation (*n* = 3).

### 3.4.3. Association of Microbial Diversity and Abundance with Environmental Variables

Our results showed increases of the alpha-diversity indices for the fungal and prokaryotic communities with composting time, Table 5. A similar change in prokaryotic biodiversity was reported by Galitskaya et al. (2017) [17]. Sun et al. (2020) [15] found that the alpha-diversity indices for prokaryotes and fungi initially increased and then decreased, reaching the lowest value in 20-day compost samples.

The relations between environmental conditions and the structure of the microbial community were assessed, Table S2. The values of the Pearson correlation coefficient (r) indicated that the carbon content and the C/N ratio in the material correlated negatively with the total number of prokaryotic and fungal OTUs estimated by the Chao1 method [43].

Therefore, with the addition of monosubstrates that increase the carbon content in the environment, we can expect a general decrease in the microbial diversity of the community.

**Table 5.** The read number and alpha diversity of prokaryotic and fungal compost communities.

| Sample Name | Number of Reads | Number of OTUs | Chao1 Index | Shannon Index * | Simpson Index * |
|---|---|---|---|---|---|
| Fungi (ITS) | | | | | |
| 0 days | n.d. | n.d. | n.d. | n.d. | n.d. |
| 7 days | 13,619 | 147 | 245 | 1.22 | 0.57 |
| 14 days | 14,757 | 221 | 439 | 2.02 | 0.18 |
| 21 days | 20,286 | 260 | 497 | 1.92 | 0.19 |
| 28 days | 12,206 | 193 | 471 | 2.46 | 0.14 |
| 56 days | 30,422 | 127 | 332 | 0.47 | 0.85 |
| 98 days | 87,190 | 472 | 713 | 2.52 | 0.17 |
| mature compost | 76,411 | 979 | 1580 | 4.12 | 0.04 |
| Prokaryotes (16S rRNA) | | | | | |
| 0 days | 17,606 | 1313 | 5934 | 2.50 | 0.34 |
| 7 days | 17,905 | 1291 | 5575 | 3.03 | 0.15 |
| 14 days | 15,900 | 1066 | 4428 | 3.58 | 0.09 |
| 21 days | 20,782 | 2012 | 6697 | 4.18 | 0.07 |
| 28 days | 18,033 | 3016 | 10,209 | 5.39 | 0.02 |
| 56 days | 15,893 | 2738 | 10,337 | 5.46 | 0.02 |
| 98 days | 16,986 | 2654 | 7176 | 5.23 | 0.04 |
| mature compost | 16,851 | 4771 | 15,385 | 6.64 | 0.01 |

* 10,000 reads normalized.

The water mass content in the substrate within the range of 36.7–72.6% also correlated negatively with the expected total OTU number and the Shannon index, while conductivity exhibited a negative correlation with the expected total number of fungal OTUs and with the Shannon index for the fungal community. Increased conductivity was caused by increasing concentrations of ammonium ions, mineral salts, and dissolved $CO_2$, which is typical during the active phase of the substrate biodegradation.

Hence, more favorable growth conditions, elevated humidity, and the presence of various ions in the medium, as well as elevated carbon content and C/N ratio, result in a significantly simpler structure of the community, with a lower total microbial diversity during periods when these conditions are maintained in the substrate. This may result from active colonization of the substrates by rapidly growing microorganisms (r-strategists), which outcompete other microbial groups. While a significant effect of humidity, C/N, and carbon content on the microbial community has been shown in a number of works [15,18,100] that reported a positive correlation with the biodiversity. The different growth environments might be the main reason for this result.

Interestingly, the germination index showed a positive correlation with the Chao1 indices for the fungal and prokaryotic communities, as well as with the Shannon index for the prokaryotic community, Table S2. The nitrification index showed a negative correlation with the Shannon index and a strong positive correlation with the Simpson index for the prokaryotic community. Therefore, the more diverse the microbial community and the more complex its structure, the higher the properties of compost that stimulate the growth and development of plants.

The abundance of the fungal genera *Penicillium* and *Aspergillus* and of the bacterial genus *Bacillus* in the community correlated positively with the substrate temperature and

carbon mineralization as $CO_2$, Table S3. Nakasaki et al. (2019) [16] also reported a positive correlation between *Bacillus* in the community and the $CO_2$ level. The abundance of the genus *Caldibacillus* correlated positively with carbon mineralization as $CO_2$. Therefore, our results showed that during the thermophilic stage bacteria of the genus *Bacillus* and fungi of the genera *Penicillium* and *Aspergillus* were the main degraders of OM from mixed FW and agricultural waste (see Table 4). A number of works also showed a significant role of these microorganisms in waste biodegradation [16,19,75]. Some authors reported the key role of *Aspergillus* during the cooling stage [101,102].

According to the correlations between microbial genera predominant during composting (see Table S4), *Aspergillus* developed together with *Penicillium*. *Penicillium* and *Bacillus* also grew together and did not suppress the growth of each other. A positive correlation was found between the abundance of the fungi *Thermomyces* and *Microascus*, which developed actively during compost maturation (see Table 4).

The growth of *Thermobifida*, which was abundant at cooling and maturation (see Table 4), exhibited a positive correlation with the development of thermophilic proteolytics and mesophilic amylolytics in the compost, Table S1. The abundance of *Thermomyces* correlated positively with the numbers of nitrogen fixers, cellulolytics, thermophilic amylolytics, and proteolytics. This fungal genus was likely a member of one or several of these physiological groups or together with them participated in compost biodegradation. The abundance of *Byssochlamys* correlated positively with development of ligninolytic micro-organisms.

The prokaryotic communities developing during the studied stages differed significantly from each other (see Figure S1). The community evolved with the changes in ambient conditions. During the mesophilic and early thermophilic stages, lactic acid bacteria (*Weisella, Leuconostoc*) were the main components of the community. Changes in the composition of prokaryotic communities during composting were more pronounced than in fungal communities, which was also shown by other authors [15,103]. According to the results obtained, the temperature was the main parameter that triggered the microbial succession. Community change under the influence of other factors was not clear.

The fungal communities that formed on days 14–28 were similar in composition; the same applied also to the prokaryotic community. On day 56, after the long-term thermophilic period, the substrate temperature decreased to 31.7 °C; at the same time, a decrease in the diversity of the fungal community was observed (Chao1 332, Shannon index 0.47, Simpson index 0.85), Table 5. At the same time, these conditions did not significantly affect the profile of the prokaryotic community.

This may be explained by the generally higher diversity and, therefore, higher resistance of the prokaryotic community (Chao1, 10,337; Shannon index, 5.46; Simpson index, 0.02). Sun et al. (2020) [15] also suggested that, during composting, the fungal community was more sensitive to temperature changes. The instability and sensitivity of the fungal compost community may have a negative effect on the rate of OM decomposition under rapidly changing conditions. The measures for increasing the diversity of the fungal community, including inoculation with thermotolerant fungal strains, seem, therefore, reasonable.

Successions of the fungal community caused by significant temperature changes likely occurred on days 7 and 56; a single genus predominated in the community, *Byssochlamys*, which survived temperature changes and displaced other members of the community.

Sun et al. (2020) [15] observed mycobiota similarity and low diversity on days 9 and 20 of composting. We observed significant differences in the microbiota composition at the stages of cooling, maturation, and mature compost for both fungal and prokaryotic communities. Sun et al. (2020) and Meng et al. (2019) [15,103] have reported opposing results, with similar communities developing during cooling and maturation.

Therefore, a dependence between community richness and biodiversity and the humidity and carbon content was shown. A relation was found between growth of the dominant genera and temperature. A positive correlation of the dominant genera with a $CO_2$ level indicates them as the major OM degraders. A comparison of prokaryotic and

fungal communities from different compost samples revealed that the fungal community was less diverse and more sensitive to temperature changes.

## 4. Conclusions

The biodiversity of microbial communities increased with the composting time, and the mature compost community was the most complex.

At the start of composting and during the transition from the mesophilic to the thermophilic stage at the high humidity conditions, lactic acid bacteria *Weisella, Leuconostoc, Limosilactobacillus* were actively growing in the community. As a result of their activity, the pH was significantly lowered, which caused a slowdown in microbial activity.

The main activity of biodegradation (according to $CO_2$ emissions) was observed during the thermophilic stage, when *Bacillus* and *Caldibacillus* were the dominant genera of bacteria, while *Penicillium* and *Aspergillus* were the dominant fungal genera.

During this period, the total microbial number significantly increased, and proteolytic, amylolytic, and cellulolytic microorganisms actively developed in the substrate. Nitrifying microorganisms also rapidly proliferated at the thermophilic stage and contributed to the reduction of the ammonium content and the accumulation of nitrates in the compost. On contrary, nitrogen-fixing and ligninolytic microorganisms developed vigorously during cooling and maturation.

High humidity and an elevated C/N ratio promoted decreased biodiversity and richness of microbial communities. During the period when such conditions existed in the substrate rapidly growing microorganisms replaced the less adapted members of the community.

The temperature had a significant effect on the species number and diversity in the fungal community, which was more sensitive to drastic temperature changes than the prokaryotic community was. *Byssochlamys,* which form temperature-resistant spores, was the main member of the fungal community during the transition from the mesophilic to the thermophilic stage and during the cooling stage. The instability of the fungal compost community may have a negative effect on the rate of organic matter decomposition in rapidly changing conditions.

It was shown that the more diverse the microbial community, the higher the properties of compost that stimulate plant growth and development, which is due to the high correlation of biodiversity with the indices of germination and nitrification.

The high initial C/N ratio did not lead to an increase in nitrogen immobilization and a decrease in the levels of plant nutrient forms. As an outcome of composting, mature compost that meets the quality criteria was obtained.

Our findings extend the understanding of the succession of microbial communities in mixed food waste and wheat straw in-vessel composting systems.

**Supplementary Materials:** The following are available online at https://www.mdpi.com/article/10.3390/agronomy11050928/s1, Table S1: Pearson correlation (r) between physiological groups and dominant genera of microbial community, Table S2: Pearson correlation (r) between community richness and diversity and physicochemical properties, Table S3: Pearson correlation (r) between dominant microbial genera and physicochemical properties, Table S4: Pearson correlation (r) between dominant microbial genera represented during composting, Figure S1: Distribution of the samples by their taxonomic composition in reduced dimensionality with the influence of ambient parameters. (PCoA): relation between carbon content, % (a); pH (b); temperature, °C (c); wetness, w% (d) and the prokaryotic (1) and fungal (2) communities throughout composting. Color of each point refers to a certain value of the parameter. The closer the samples (points) on the plot, the more similar their composition; vectors show the directions in which the levels of the respective major taxa increased.

**Author Contributions:** Conceptualization, V.M.; methodology, V.M., A.V. and A.M.; software, A.M.; validation, V.M. and A.M.; formal analysis, V.M., A.V. and A.M.; investigation, V.M., A.V. and A.M.; resources, V.M. and A.M.; data curation, V.M. and A.V.; writing—original draft preparation, V.M., A.V. and A.M.; writing—review and editing, V.M., A.V. and A.M.; visualization, V.M., A.V. and A.M.;

supervision, V.M.; project administration, V.M.; funding acquisition, V.M. All authors have read and agreed to the published version of the manuscript.

**Funding:** The article was made with the support of the Ministry of Science and Higher Education of the Russian Federation in accordance with agreement No 075-15-2020-907 on 16 November 2020, providing a grant in the form of subsidies from the Federal budget of the Russian Federation. The grant was provided for state support for the creation and development of a world-class scientific center "Agrotechnologies for the Future".

**Conflicts of Interest:** The authors declare no conflict of interest. The funders had no role in the design of the study; in the collection, analyses, or interpretation of data; in the writing of the manuscript, or in the decision to publish the results.

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
