# Peer review of "Microbiological Activity during Co-Composting of Food and Agricultural Waste for Soil Amendment"

_agronomy, doi:10.3390/agronomy11050928_

Round 1

Reviewer 1 Report

This version of the manuscript is very much improved and I think I can see that the work is worth publishing now. However, there still need to be major revisions to make it more readily understandable.

It is still very long with too many figures - 14 in the manuscript plus supplementary results.  I also became confused with some of the conclusions drawn (for example biodiversity increases at line 865 and decreases at line 870) and so I think you need to make the manuscript clearer.

The strength of this work is that you have both detailed physicochemical data and detailed microbiological data yet the discussion of how these are linked is the shortest section and hidden away at the end of the results and discussion section in Table 4 and section 3.4.3. This is, to my mind, the most important and interesting part of the work. Consider re-structuring section 3 to make this more prominent and actually drawing some conclusions to include in section 4.

Specific comments:

Abstract “causing a decrease in the richness and biodiversity of the microbiota”.  This is plainly not true as the biodiversity of the final mature compost has actually increased. I suspect that you mean in the thermophilic stage – please correct this so that it is clear.

Methods section is well written and clear.

Results and Discussion.

This section is very long and rambling. I suggest sub-headings that are more informative and splitting the first section into more sub-sections? Section 3.1 Dynamics of the physicochemical parameters… is 6.5 pages long.  Some more headings within that would be helpful. The microbiological section is broken up into smaller sections: “total numbers”, “various physiological groups” and “as determined by NGS profiling”. It would be better to use headings that describe the results rather than the methods.

Figure 3 (and elsewhere throughout manuscript) I have never seen moisture content referred to as “% w”. Please correct

It is very difficult to follow the relationship between the temperature, mixing, water addition and aeration across all of the different figures on different pages. It would be helpful to either combine figures 2 – 7 (perhaps as panels A – F on one figure) or to overlay the temperature changes on all the figures so that the reader can see, for example, where the thermophilic phase is in relation to CO2 generation.

One of the aims of the study was to study the relationship between the microbial community and the ambient conditions and this would be a lot easier to assess if the data was presented together in combined figures or at least in fewer figures. As above, presenting the different physiological groups as different panels on one figure and adding in something to make it clear where temperature changed, and mixing and water addition occurred would make it a lot easier to interpret.

Line 251-252 “a shift in the composition of the microbial community likely occurred” You actually have this data on community composition – was a shift in the community composition observed in your sequencing or plate count data?

Line 257 How do you know that the reaction rates were higher?

Line 286 I don’t see “wave like” changes in CO2 emission – I am not sure what is meant by this. How do you know that the changes in CO2 emission are due to moisture evaporation and stirring? This would be a lot easier to judge if the data was presented within the same figure.

Line 339 “Nitrifying microorganisms likely contributed” You discuss the various forms of nitrogen detected here but don’t refer to your microbiological results at all! What evidence do you have from your microbiological studies that this is actually important – you have both plate count data and sequencing data so should be able to provide some evidence that they either did or did not contribute.

Line 811 “The shares of the fungal genera” – what does this mean? Re-word this sentence.

In general, the results section would benefit from discussing, where relevant, the physicochemical and microbiological data together to answer aim ii. There is a little bit of this in section 3.4.3 but it doesn’t really combine the abundance data of various physicological groups with the microbiome sequencing data very well.

The conclusion section is too brief and does not provide conclusions to the aims that were stated in the introduction. Please consider expanding to explicitly describe what conclusions you drew in relation to the three aims. Given how long the rest of the manuscript is, a conclusion section of 12 lines is not adequate.

Reviewer 2 Report

Comments and suggestions

  1. I propose to supplement the caption of figure 2 with data on the times of stirring stages, which were marked with numbers from 1-5
  2. Fig. 8 is illegible, I propose to divide it into parts a and b for mesophilic and thermophilic organisms, respectively. You can proceed in a similar way from fig. 10.
  3. Figures 11 and 12 in their current form are unacceptable. Please work through them in such a way that it is possible to trace the course of variability. They can be divided separately into mesophilic with samples collected in 3 terms, mesophilic - with 4 terms. A similar procedure can be applied to thermophilic ones. Other solutions can be used to clearly present the data. Please note that each figure has one legend, it is not repeated separately for figures a and b, if they appear under one description. The coordinate system in drawings does not necessarily start with 0.
  4. Another ordering rule applies to the order of the description according to one scheme, for example the description is mesophilic then thermophilic. Unless there is some uniqueness, which I do not see in this case, page 16, opening line 520.
  5. An in-text explanation for stage I and II, page 17, line 547 is needed.

General comments

My substantive comments made during 1 review were taken into account by the authors.

The current proposals presented relate to the presentation of data.

Round 2

Reviewer 1 Report

Thank your for further revising your manuscript. It is now much clearer and easier to follow.  The conclusion section in particular now demonstrates the significant improvements to our understanding of the composting process.